# Geometry image super-resolution with AnisoCBConvNet architecture for efficient cloth modeling

Jong-Hyun Kim[1], Sun-Jeong Kim[2], Jung Lee[2]*

**1** School of Software Application, Kangnam University, Yongin, Gyeonggi, Republic of Korea, **2** School of Software, Hallym University, Chuncheon, Gangwon, Republic of Korea

* airjung@hallym.ac.kr

**Data Availability Statement:** All relevant data are within the manuscript and its Supporting information files.

**Funding:** This research was supported by Basic Science Research Program through the National

## Abstract

We propose an anisotropic constrained-boundary convolutional neural networks (hereafter, AnisoCBConvNet) that can stably express high-quality meshes without oscillation by applying super-resolution operations to low-resolution cloth meshes. As a training set for the neural network, we use a pair between simulation data of low resolution (LR) cloth and data obtained by applying the same simulation to high resolution (HR) cloth with increased quad mesh resolution of LR cloth. The actual data used for training are 2D geometry images converted from 3D meshes. The proposed AnisoCBConvNet is used to train an image synthesizer that converts LR geometry images to HR geometry images. In particular, by controlling the weights anisotropically near the boundary, the problem of surface wrinkling caused by oscillation is alleviated. When the HR geometry image obtained through AnisoCBConvNet is converted back to the HR cloth mesh, details including wrinkles are expressed better than the input cloth mesh. In addition, our results improved the noise problem in the existing geometry image approach. We tested AnisoCBConvNet-based super-resolution in various simulation scenarios, and confirmed stable and efficient performance in most of the results. By using our method, it will be possible to effectively produce CG VFX created using high-quality cloth simulation in games and movies.

## Introduction

Cloth simulation, which is one of many fields of physics-based simulation, is recently widely used in various fields such as VFX (Visual special effects) used in movies and animation, CF, and virtual fashion show based on VR/AR. The realistic expression of the folding patterns that appear when the cloth is folded, one of the characteristics of the high-quality cloth model, is very important in expressing the style of the virtual character and its unique animation [1–4]. Physics-based simulation can numerically calculate realistic and detailed cloth deformation, but requires complicated numerical analysis and high computational cost. One of the common approaches to speed up HR calculations in computer graphics is to find the appropriate LR space in the process of capturing the entire motion and map it to the HR domain. These

Research Foundation of Korea(NRF) funded by the Ministry of Education(2022R1F1A1063180, 2020R1A2C10151953). This research was supported by a Hallym University Research Fund (HRF-202108-002). This study was carried out with the support of ´R\&D Program for Forest Science Technology (Project No. 2021390A00-2123-0105)´ provided by Korea Forest Service (Korea Forestry Promotion Institute).

**Competing interests:** The authors have declared that no competing interests exist.

approaches typically uses precomputed data and data-driven techniques [5, 6]. This method has been applied to represent wrinkles in cloth animations [7, 8], and has been extended to various algorithms such as subspace simulation method [9] and pose space deformation method [10].

Another approach to improve the detail of cloth simulation is SR (super-resolution) technique that tunes LR mesh to HR mesh. SR has been continuously studied in computer vision and graphics fields to generate HR images from LR images [11, 12]. This method has recently made great progress with the advent of techniques such as Deep ConvNet (Convolutional Neural Networks) and GAN (Generative Adversarial Networks) [13–15]. ConvNet is a powerful machine learning tool and is very useful for data-driven applications such as image style transfer [16], speech synthesis [17], and natural language processing [18, 19]. It is not easy to deal with irregularly structured 3D mesh data due to the nature of ConvNet, which mainly deals with the data structure of 2D array. Gu et al. proposed a geometry image technique that converts a 3D mesh into a 2D image to handle surface parametrization [20]. We thought that if we could convert the mesh to image or from image back to mesh using this technique (Fig 1A and 1B), we could apply ConvNet-based image SR. We first convert LR mesh to LR geometry image, then enhance it to HR image using ConvNet, and then convert it back to HR mesh. The 3D coordinates of the vertices constituting the cloth mesh are converted to RGB and stored in the geometry image in the form of a 2D array. "Problem Statement" section describes some issues that appear when training a cloth model through ANN (Artificial Neural Networks).

## Problem statement

In this study, ConvNet, a sub-concept of ANN, will be used, and in this section, we will explore what issues arise when ANN is used for physics-based simulations. Recently, advanced studies

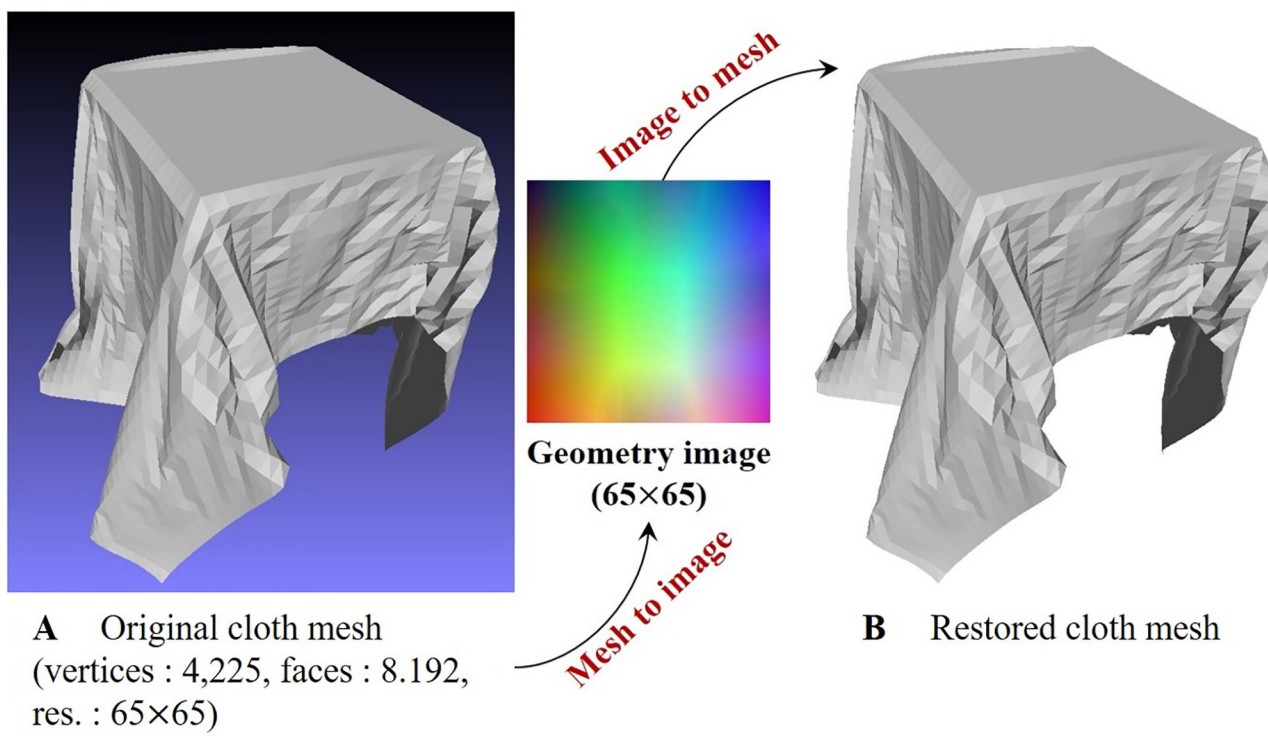

**A** Original cloth mesh
(vertices : 4,225, faces : 8.192, res. : 65×65)

**Geometry image (65×65)**

**B** Restored cloth mesh

**Fig 1. Visualizing geomety image from 3D cloth mesh.**

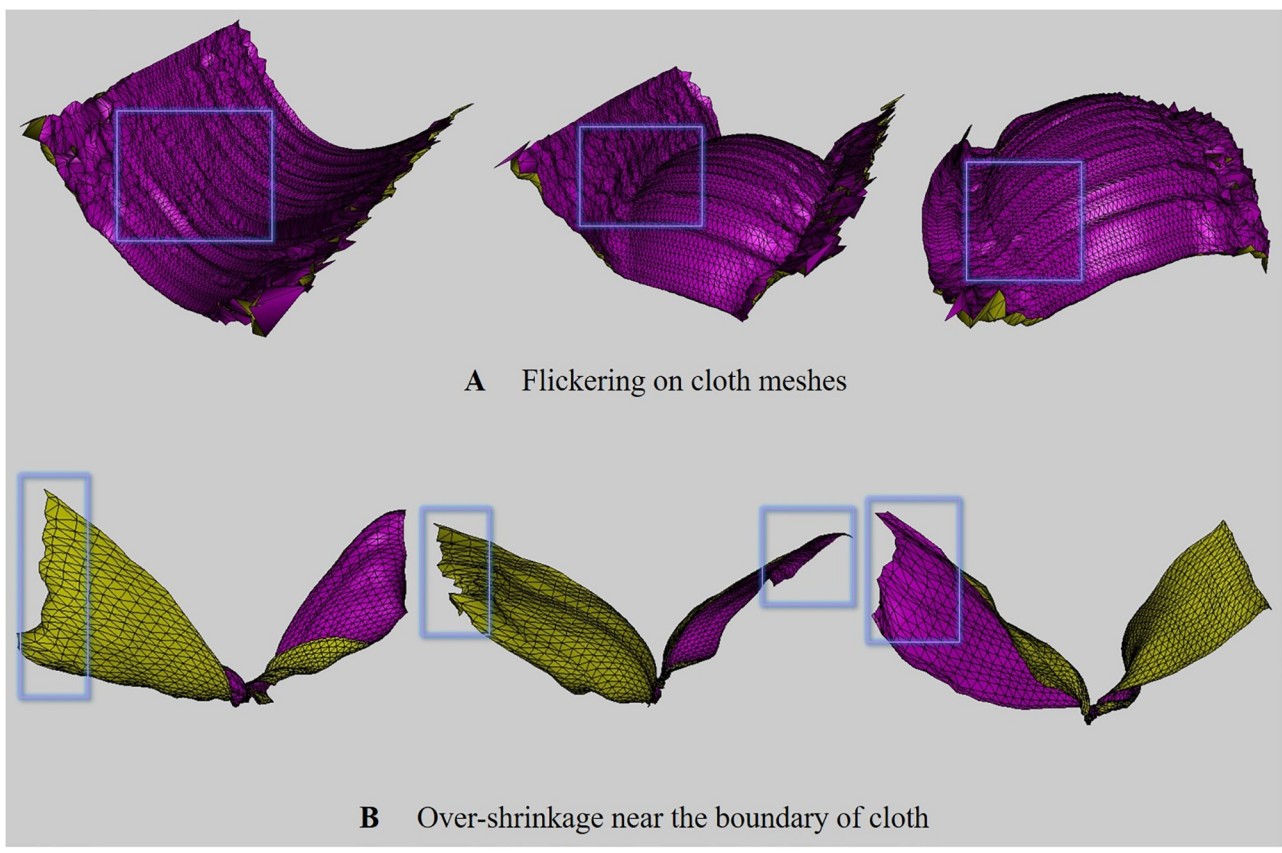

**A** Flickering on cloth meshes

**B** Over-shrinkage near the boundary of cloth

**Fig 2. Issues in upscaling cloth meshes with ANN [27, 28] (input cloth res.: 32 × 10, output cloth res.: 64 × 20, box: Awkward geometry).**

have been introduced to improve the efficiency and details of physics-based simulations based on ANN. In the field of fluid simulation, attempts have been made to improve the efficiency by replacing the large amount of calculation of the Poisson equation, which must be solved when calculating the pressure of HR simulations, with a light SR operation [21–26]. However, apart from fluid simulation, the process of upsampling the geometry of fluid surfaces has more sensitive issues. In simulation using volumetric density, one outlier value does not have a big influence on the result. However, when converting 3D objects to 2D geometry images, if a single pixel value is incorrectly sampled, it affects the vertex positions of objects, so position flickering or over-shrinkage near boundary surfaces occurs (Fig 2).

**Flickering problem.** Recently, Chen et al. proposed a technique for synthesizing cloth wrinkles based on ConvNet [27, 28]. Similar to our method, this approach converts 3D objects to geometry images to model cloth surfaces. However, this method has two major limitations: 1) It does not work properly in the ANN commonly used when performing image SR, but only properly in SRResNet, as mentioned in their paper. 2) In the process of converting geometry image back to 3D object, if the cloth moves rapidly like a twisting motion, noise is included in the vertex position, causing flickering that distorts the surfaces (Fig 2A). This problem becomes more serious when using a network model other than SRResNet [29].

**Over-shrinkaged problem.** Recently, Oh et al. introduced a technique to perform cloth simulation hierarchically using DNN (Deep Neural Networks) [30]. In this approach, a coarse level of cloth simulation is performed using traditional physically-based simulation, and a more detailed level is generated by inference using DNN models. The difference from Chen

et al.'s technique [27] is that they still use the vertex coordinates of 3D objects, not 2D geometry images. Similar to the loop subdivision method that is often used in geometry processing, one triangle is subdivided into four, resulting in the same level of results as HR cloth simulation. However, as mentioned in their paper, this technique caused over-shrinkage near boundary surfaces as in Chen et al.'s technique [27] due to inaccurate inferences of DNN. As shown in Fig 2B, distortion can be seen near both sides of the cloth surfaces when the cloth surfaces are strongly deformed. Although the initial cloth model of this scene is a rectangle mesh, the surfaces near the boundary were over-shrinked during the cloth twisting process.

## Related work

In this section, we briefly explore some techniques closely related to our research, data-driven cloth modeling, image SR using ANN, and ANN techniques used in 3D shapes and simulations.

### Data-driven cloth modeling

Data-driven methods are widely used because they can produce cloth animation quickly. These methods are broadly classified into two groups.

The first group simulates on a coarse mesh using precomputed data to add geometric datails. Feng et al. [31] introduced a method for decomposition of HR details in mid or fine scale deformation. Since mid or fine-scale details are extracted from coarse-scale simulation, it shows fast performance enough to be performed in real time. Wang et al. [8] presented an example-based approach that can reinforce the details of coarse simulations using the wrinkle database obtained from numerous HR simulations. Kavan et al. [32] introduced a method to improve the details of cloth simulation by training linear upsampling operators from numerous HR simulations. Zurdo et al. [10] proposed an example-based technique to augment the details of coarse simulations by combining multi-resolution and pose space deformation (PSD) techniques. Hahn et al. [9] proposed a method for performing subspace simulation using a low-dimensional linear subspace with temporally adaptive properties. In this method, full-space simulation training data was used to construct a pool of low-dimensional bases distributed in the pose space.

The second group consists of deformation approaches using precomputed data to avoid runtime simulations. De Aguiar et al. [33] proposed a method for learning a linear conditional cloth model using data obtained from physics-based simulations. Although this method performs very quickly, it is not sufficient for various scenes because it aims for simple cloth simulations with little folding. Guan et al. [34] proposed a technique for cloth deformation from body shape and pose, and trained a linear model for rapidly deforming cloth according to various body shapes and poses without runtime simulations. Kim et al. [35] performed cloth animation quickly by constructing a secondary motion set using the input primary motion graph. Kim and Vendrovsky [36] expressed cloth deformation by using the animation data that the character wears as precomputed data.

Holden et al. recently suggested a technique for processing interaction with external objects effectively by combining subspace simulation with machine learning [37]. This approach efficiently and stably expresses simulation deformation effects such as external force and collision. Because this method is concerned with simulation deformation caused by interaction (e.g., self-collision, interactions with exterior objects, etc.), it is distinct from performing SR operations in cloth simulation. However, we believe it might be incorporated for future detailed enhancement.

Wang et al. recently presented a method for semi-automatic garment authoring animations based on deep learning [38]. This method implemented a potential garment representation for motion-independent intrinsic parameters (e.g., gravity, cloth material, etc.). Zhang et al. presented a framework for producing a realistic dynamic garment image sequence, taking into account the movement of the body joints [39]. Given the avatar's joint motion sequence, this method generates a plausible dynamic garment form even at the point of blind spot. Chen et al. proposed a novel framework for synthesizing high-resolution cloth dynamics in low-resolution meshes [40]. When mapping from coarse to fine meshes, this approach conducts large-scale deformation. Zhang et al. suggested a data-driven method for improving the detail in coarse garment geometry. This method expressed high-resolution details by matching Gram matrix based on style loss [41]. Most solutions do not directly simulate cloth (dynamics and collection handling), but rather synthesize virtual garment based on the avatar's movement or form. It is important to consider the cloth material or external force throughout this process. Also, there are studies focusing on data enhancement, and our method is one of them.

## Image super-resolution

Although it is difficult to apply SR to a single image without prior information, ConvNet or GAN-based methods have recently made great progress using sufficient training data. Dong et al. [12] used the bicubic interpolation data of LR images as input and used a simple 3-layers ConvNet to generate HR images. Kim et al. [42] proposed a DRCN (Deeply-Recursive Convolutional Networks) technique, which improved the performance of ConvNet by reducing the number of parameters using a recursive structure with a depth of 20 layers. In order to speed up the calculation and add more layers, many studies do not use HR images to which bicubic interpolation is applied as input, but use LR images to upscale the feature map to HR in the last few layers of the network. For example, fast SR ConvNet [43] uses transpose convolutional layers called deconvolutional layers, and efficient sub-pixel ConvNet [44] uses sub-pixel convolutional layers to solve the upscale problem. In the SR field, equality evaluation of algorithms is an important problem, and their optimization target is to minimize the mean squared error (MSE) between the ground truth and the recovered HR image.

Recently, Mei et al. presented image SR technique using cross-scale non-local attachment and exhaustive self-example mining [45]. Most image SRs perform learning process in large-scale external image resources for local recovery. In this process, most existing methods ignore the long-range feature-wise similarity of images, and this study suggests a solution to this problem. As a result, this study can efficiently process SR in natural images. This technique, however, is not ideal for 3D geometry-based upsampling, which is the purpose of our research, because the boundary of the cloth surfaces is distorted and flickering occurs during the conversion of the 2D pixel to 3D position. Song et al. suggested AdderNets method to improve energy efficiency of image SR process [46]. By calculating the output function using addition, this method minimizes the energy usage of the multiplication operation. Applying the image classification technique to the image SR is challenging. Specifically, the adder operation has difficulty learning the identity mapping required for image processing, but this study suggests a solution. We expect that integrating AdderNets into our methods will improve its efficiency, but it is difficult to effectively express cloth SR using AdderNets only.

## Convolutional neural networks for 3D

Compared to 2D images, 3D shapes are relatively difficult to process in ConvNet due to their irregular connectivity. Nevertheless, several related studies have been conducted in various fields in recent years. Su et al. [47] proposed a technique to express 3D shapes using multi-

view projections and panoramic views. Wu et al. [48] proposed a technique for voxelization of 3D shapes using DBN (Deep Belief Networks). Girdhar et al. [49] proposed a technique to reconstruct 3D shapes from 2D inputs by combining an encoder for 2D images and a decoder for 3D models. Yan et al. [50] created 3D models from 2D images by adding projection layers to convert 3D to 2D. Choy et al. [51] proposed novel recurrent networks for mapping 3D shapes from images of objects. Li et al. [52] and Nash and Williams [53] proposed a new ANN for encoding and synthesizing 3D shapes using pre-segmented data. Chu and Thuerey [25] synthesized HR smoke by encoding the similarities between LR and HR fluid patches based on ConvNet for animation production. Since then, many techniques have been used in the fluid simulation field [21–24]. In this paper, we introduce a technique to anisotropically constrain the boundary of cloth surfaces and improve LR cloth meshes with HR details using ConvNet.

## Our framework

In this paper, by performing SR on the 2d geometry image obtained by projecting 3D cloth meshes into image space, it is quickly upscaled to a high-quality geometry image. PBD (Position based dynamics) [54] was used as a dynamics solver to obtain cloth surfaces data, and since meshes obtained by simulation are used as input data at runtime stage, cloth surfaces data created in various scenes can be utilized. Our algorithm operates as follows (Fig 3A):

1. After performing cloth simulation using PBD, the geometry image $\delta$ is created by converting the vertex positions $[x, y, z]$ of cloth meshes to $[r, g, b]$.

2. Geometry images are upscaled using AnisoCBConvNet-based synthesizer (Fig 3B). To reduce the noisy surfaces and shrunk boundaries when learning cloth anmations, we propose an architecture of three networks.

  • GeometryNet: Instead of using the color of the geometry images, we train the SR operation using smoothed residual images, the difference between upsampling and downsampling.

  • BoundaryNet: Constraint conditions are added to alleviate the noisy distortion that occurs near the boundary of the cloth.

  • EnhanceNet: To emphasize the wrinkling of the cloth, constraint conditions are added based on the contrast edge map of the cloth.

3. 3D cloth meshes are converted from upscaled 2D geometry images.

## Conversion between cloth meshes and geometry images

Modeling cloth based on geometry images has a wide field of application because it can be easily trained and tested without complex numerical solutions or in-house solutions. In some approaches, since there is no continuous connection information like geometry images because public image datasets are used, flickering often occurs when performing SR, and additional network models are used to mitigate this problem [27, 28]. The proposed method is not subject to these limitations, and in the training stage, not only the CIFAR-10 and COCO datasets but also the geometry images of the cloth surfaces produced by the physics-based simulator were used. PBD was used as the cloth solver, and two functions required in the training and test phases are calculated as follows: 1) function $\gamma_{c \to i}$ to convert cloth meshes to geometry

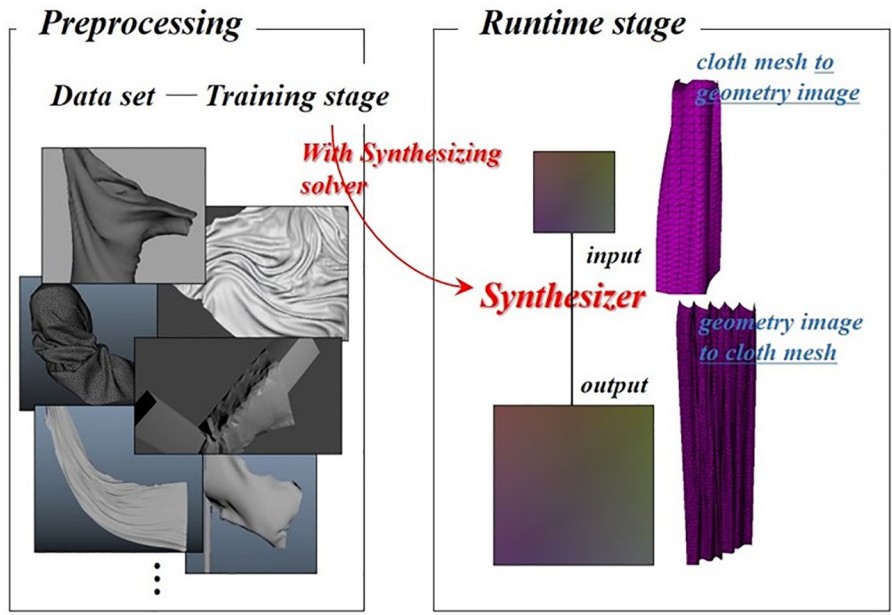

**A    Algorithm overview**

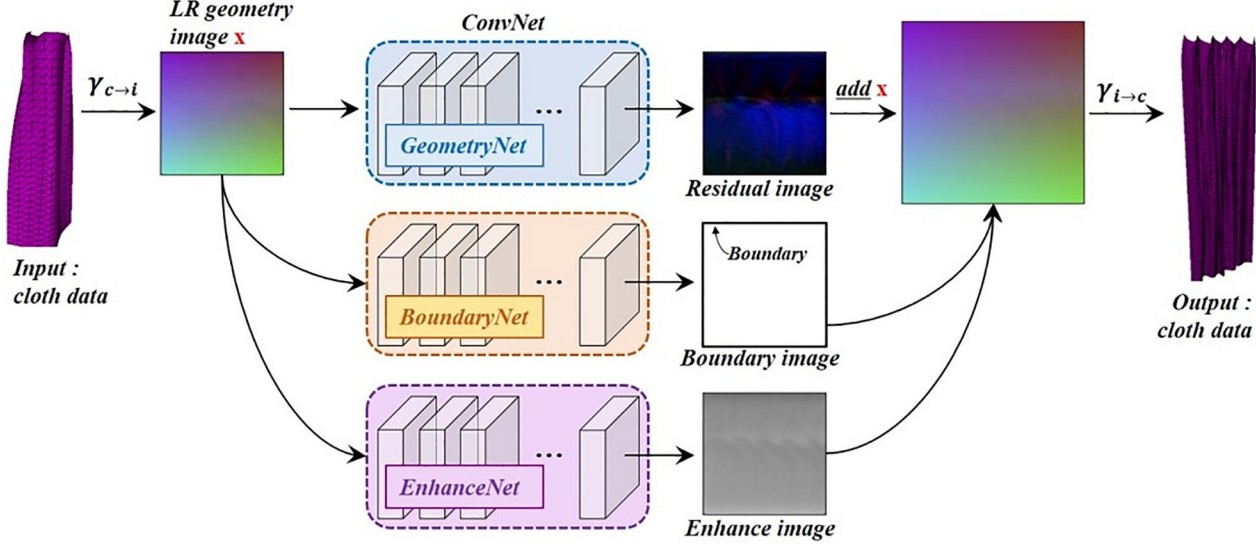

**B    Synthesizer based on AnisoCBConvNet(Anisotropic Constrained-Boundary ConvNet with 3 types of constraint networks)**

**Fig 3. An overview of our algorithm.**

images, and 2) function $\gamma_{i \rightarrow c}$ that does the opposite (Eqs 1 and 2).

$$\gamma_{c \rightarrow i} = \frac{p - b_{min}}{\|b_{max} - b_{min}\|} \qquad (1)$$

$$\gamma_{i \rightarrow c} = c_i^{rgb} \|b_{max} - b_{min}\| + b_{min} \qquad (2)$$

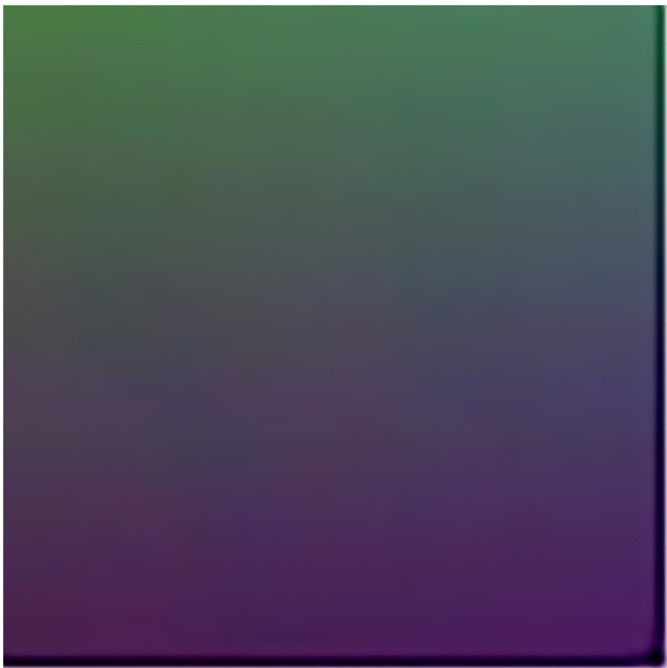

**Fig 4. Geometry image with $\gamma_{c \to i}$.**

where $p$, $b_{min}$, and $b_{max}$ indicate the vertex position of the cloth mesh and the position of the minimum/maximum value of the simulation domain, respectively, and $c_i^{rgb}$ is $[r, g, b]$, the color converted from $[x, y, z]$. If we convert all vertices of cloth mesh into RGB space using Equation and visualize them as images, we can get color results that change smoothly like color gradation (Fig 4). Since this result is simulated within domain $b_{min,max}$, the position of the vertex is clamped between 0 and 1, and Fig 4 shows the result of multiplying each component of the converted position by 255 and expressing it in color. The color is expressed as an integer only for visualization, and the float type is used in the actual calculation process.

The process of converting a geometry image to cloth meshes is conceptually $\gamma_{c \to i}^{-1}$, but we simply convert it using Eq 2. Fig 1 shows the process of converting a 3D cloth mesh into a 2D geometry image using Eq 1 and restoring it again through Eq 2. In this process, when converting two different spaces, it is possible to convert without precision error because floating type is used. Fig 5 shows the correctly restored mesh without error when converting the $64 \times 20$ resolution geometry image to cloth mesh. In this section, the structure that converts between the cloth mesh and the geometry image is explained, and in the next section, the network models for SR of the converted geometry image are explained.

## AnisoCBConvNet(Anisotropic constrained-boundary convnet)

The architecture proposed in this paper is divided into three subnetworks: They are a GeometryNet that performs SR by converting the vertex position of the cloth to $[r, g, b]$, a BoundaryNet that alleviates the problem of over-shrinking due to noise generated near the boundary, and an EnhanceNet that emphasizes the wrinkling effect of the cloth based on edges. These multiple outputs are connected and used to obtain the final resulting geometry image.

**GeometryNet for smoothing surfaces.** After obtaining the LR cloth meshes set $\{O_l^0, O_l^1, \ldots\}$ and the HR meshes set $\{O_h^0, O_h^1, \ldots\}$ by physics-based simulation, the LR

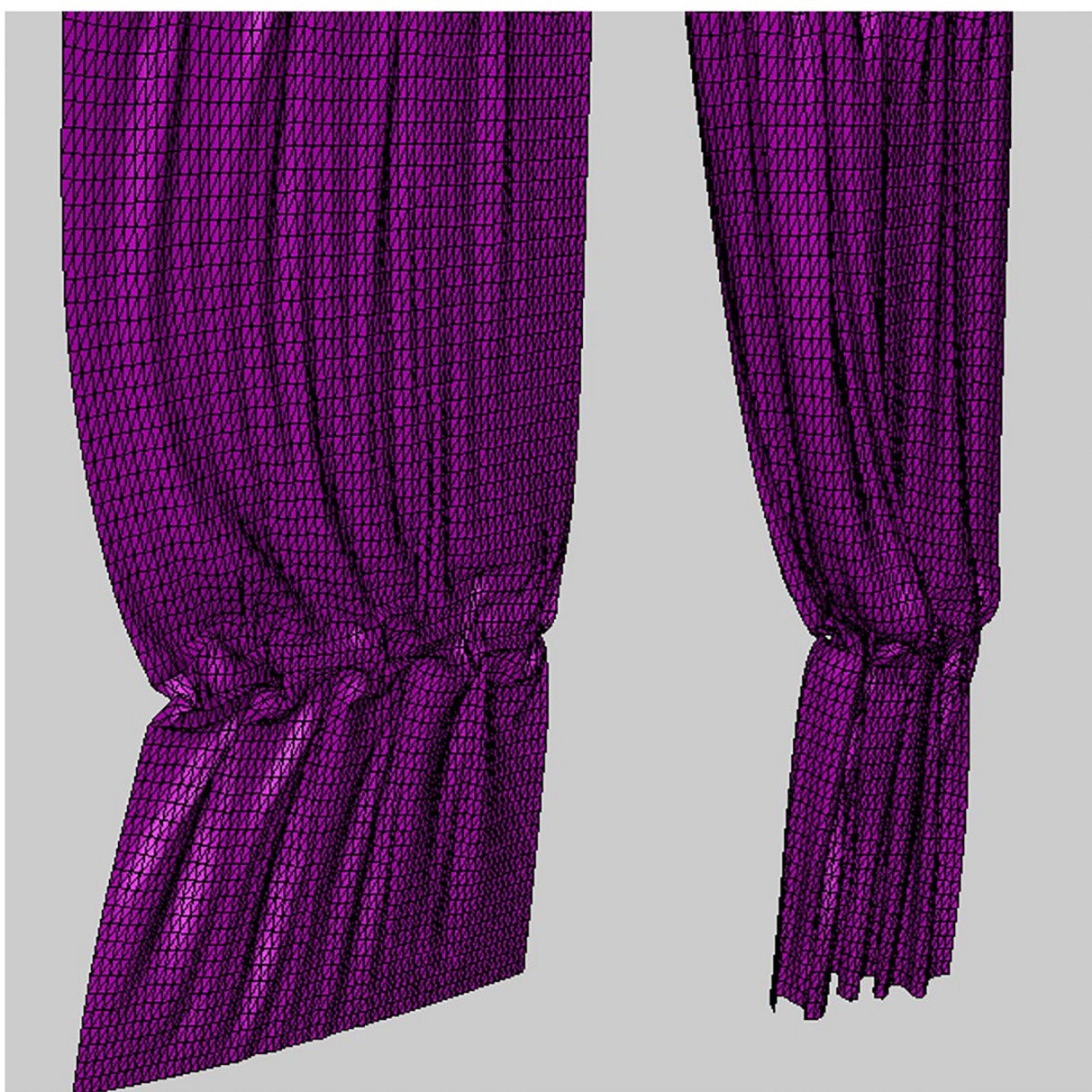

**Fig 5. Restored cloth with $\gamma_{i \to c}$.**

geometry image $\{\delta_l^0, \delta_l^1, \ldots\}$ and HR geometry image $\{\delta_h^0, \delta_h^1, \ldots\}$ are generated using the method described in the previous section, respectively. Each geometry image is split into patches before being fed into the training networks. Given the training data, our goal is to find a mapping function $f(\mathbf{x})$ that minimizes the loss between the predict values $\delta_s$ and the ground truth $\delta_h$. The object function for performing this process is the MSE between the predicted image and the ground-truth image. Our goal is to train a model $f$ that predicts the $\delta_s = f(\mathbf{x})$ value, and consequently to minimize $\frac{1}{2} \|\delta_h - f(\mathbf{x})\|^2$, the MSE for the training set.

In the classic SRCNN technique [12], the network must preserve all input detail since the image is discarded and the output is generated from the learned features alone. In addition, when using many weight layers, very long-term memory is required, and when SR is extended

to dynamics fields such as cloth or fluid simulations, even one pixel incorrectly positioned on the boundary and surfaces can cause serious noise, resulting in flickering and shrinkage. To alleviate these problems, we solve this problem through residual learning based on anisotropic constrained-boundary. Residual images of input/output geometry images are calculated as follows: $\mathbf{r} = \delta_h - \delta_l$. The loss function in the SRCNN method is $\frac{1}{2}\left\|\delta_h - f(\mathbf{x})\right\|^2$, but since we want to predict the residual images, the final loss function $L_G$ of GeometryNet is expressed as Eq 3

$$L_G(r, x) = \frac{1}{2}\left\|r - x\right\|^2 \tag{3}$$

where $r$ is the residual, and $x$ is the value of $f(\mathbf{x})$. In networks, loss layers are calculated using three components: residual estimate, LR geometry image, and HR geometry image. Loss is the Euclidean distance between the reconstructed image and the HR geometry image, where the reconstructed image is the sum of the network input and output images.

**Anisotropic BoundaryNet for boundary correction.** In approaches using ANN in dynamics-based simulation techniques, various types of noise often appear in common. Chu and Thuerey [25] applied ANN to smoke simulation, but flickering problem occurred because continuity was not satisfied in grid structure. Xie et al. [24] tried to solve this problem using GAN, but could not come up with a complete solution. They used overlapping grids to reduce noise appearing in the interface section between grids. This problem appears more clearly when it is a mesh-based structure rather than a volumetric structure. This technique is not widely used in the mesh infrastructure because it affects vertex position if the value is incorrectly assigned even at one node, and Chen et al. [27, 28] and Oh et al. [30] have mentioned this issue. Chen et al. [27, 28] tried to alleviate this problem by adding some padding corresponding to extra rows or columns to the boundary region of the image. However, when the boundary area is set to zero padding, noise appears near the boundary surfaces or the surfaces shrink when restoring to cloth meshes. They tried forcibly mirroring the boundary pixels, but it still didn't work. As a result, they simply copied the boundary pixels several times, and selected the best result from them. As can be seen from Chen et al.'s technique [27, 28], most of the results are simple and the movement of the cloth is limited. If the movement of the cloth is large, distortion appears in the duplicated boundary pixels (Fig 2B).

We use BoundaryNet to solve this problem. Boundary map $\delta_b$ is a binary image, and the value of each pixel is 1 if it belongs to a boundary, and 0 otherwise. BoundaryNet classifies and labels boundary vertices calculated from 3D objects. BoundaryNet is trained to estimate the boundary-map from the input image to be as close as possible to the ground-truth boundary-map obtained by applying the boundary detector to the ground-truth image. We compute the boundary-map by analyzing the distribution of vertices in cloth meshes using anisotropic kernel. We calculate the orientation of the vertices using a weighted-average-based covariance matrix $C_i$ (Eq 4).

$$C_i = \frac{\sum_{j \in N_2}(p_j - \bar{p}_i)\sum_{j \in N_2}(p_j - \bar{p}_i)^T W(p_j - \bar{p}_i, d)}{\sum_{j \in N_2} W(p_j - \bar{p}_i, d)} L(p_i) \tag{4}$$

$$\bar{p}_i = \frac{\sum_{j \in N_2} p_j W(p_j - p_i, d)}{\sum_{j \in N_2} W(p_j - p_i, d)} \tag{5}$$

$$W(\mathbf{r}, h) = \begin{cases} 1 - \left\|\mathbf{r}\right\|^2/h^2, & 0 \le \left\|\mathbf{r}\right\| \le h, \\ 0, & \text{otherwise.} \end{cases} \tag{6}$$

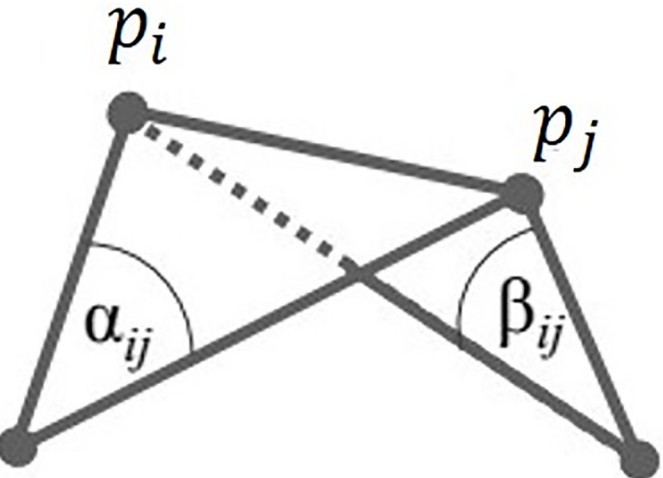

**Fig 6. Cotangent weight.**

where $d$ is a value that is $\frac{1}{10}$ of the longest length among the edges of the initial cloth mesh, and $\bar{p}_i$ is the position using Laplacian smoothing (Eq 5). $W$ is the isotropic weighting kernel (Eq 6), and $L(p_i)$ is the Laplacian operator using cotangent weights (Eq 7 and Fig 6).

$$L(p_i) = \frac{1}{\sum_{j \in N_2} W_{ij}} \left( \sum_{j \in N_2} W_{ij} p_j \right) - p_i \tag{7}$$

$$W_{ij} = \frac{1}{2} \left( \cot \alpha_{ij} + \cot \beta_{ij} \right) \tag{8}$$

The covariance matrix $C_i$ calculated by the above Equations is used to obtain eigenvectors and eigenvalues using SVD (Singular value decomposition) (Eq 9).

$$C_i = \begin{bmatrix} e_1 & e_2 & e_3 \end{bmatrix} \begin{bmatrix} \sigma_1 & & \\ & \sigma_2 & \\ & & \sigma_3 \end{bmatrix} \begin{bmatrix} e_1^T \\ e_2^T \\ e_3^T \end{bmatrix}, \tag{9}$$

where $e_n$ is the principle axes ordered by variance, and $\sigma_n$ is the stretch. The following condition was used to find the boundary vertices: $\sigma_3 \le \gamma \sigma_1$, where $\gamma$ is a threshold that determines the size of the pointed shape. Also, the vertices of the boundary edge with one adjacent triangles sharing the edge were classified as boundary vertices. As mentioned earlier, the classified vertices are converted into geometry images and used for training. The ground truth boundary-map is represented by $\delta_b^{gt}$, and the cross entropy loss is formulated as Eq 10.

$$L_B(\delta_b) = -\frac{1}{N} \sum_{x,y} \left( \delta_b^{gt}(x, y) \log(\delta_b(x, y)) \right)$$

$$+ (1 - \delta_b^{gt}(x, y)) \log(1 - \delta_b(x, y)) \tag{10}$$

where $N$ is the number of pixels in the boundary-map.

**EnhanceNet for wrinkling effects.** EnhanceNet aims to obtain a result similar to the ground-truth edge-map created by applying the Canny edge detector to the ground-truth image, and as a result, it is trained to create an edge-map from the input image. A better method can be used to reconstruct small details, but in this paper, we simply used the Canny edge detector. Enhance map $\delta_e$ is a binary image, and its value is 1 when each pixel belongs to an edge, and 0 otherwise. The networks of EnhanceNet are similar to those of BoundaryNet. The ground truth enhance-map is $\delta_e^{gt}$, and the cross entropy loss is formulated as Eq 11.

$$
L_E(\delta_e) = -\frac{1}{N}\sum_{x,y}\left(\delta_e^{gt}(x,y)\log(\delta_e(x,y))\right)
$$
$$
+(1-\delta_e^{gt}(x,y))\log(1-\delta_e(x,y))
$$
(11)

where $N$ is the number of pixels in the enhance-map. Existing approaches produce distortion on the cloth surfaces when the movement of the cloth is large, but we alleviated the noise problems and surface shrinkage near the boundary by using the anisotropic kernel described earlier.

## Composition of feature maps

In this paper, we train geometry images of cloth using three networks. The final 3D cloth meshes are restored using the three feature maps extracted in these processes. The result of GeometryNet is converted into 3D space using $\gamma_{i\,\rightarrow\,c}$. In computer vision, feature maps are trained once more and used again, but in this paper, three maps are composited using constraints (Eq 12).

$$
\Gamma_{final} = \underbrace{\Gamma^g + \eta(\Gamma^g\Gamma^e)}_{\text{enhancement of geo. img}} + \underbrace{\delta_h\Gamma^b}_{\text{boundary corr.}}
$$
(12)

where $\Gamma_{final}$ is the final cloth geometry image, and $\delta^h$ is the HR geometry image. Superscript g, b, and e refer to the resulting image obtained through GeometryNet, BoundaryNet, and EnhanceNet, respectively. The first term is the process of enhancing the geometry image, and the size of filtering $\Gamma^g$ is affected by $\Gamma^e$: The value obtained by $\Gamma^g\Gamma^e$ is a feature vertex classified by the edge detector and can be easily controlled by the user using $\eta$, an enhancement factor. The second term is the process of correcting the boundary and uses $\Gamma^b$: The value obtained by $\delta_h\Gamma^b$ is the boundary vertex classified due to the cotangent weight and the anisotropic kernel, and this value is multiplied by $\delta_h$. Since the boundary distortion problem appears in the network, we used $\delta_h$, and since only the boundary vertices remain in the multiplication process, the distortion problem can be alleviated stably.

When transforming cloth meshes of non-rectangular shape, parametrization is required, and there are several methods for doing this. Using the ARAP (As-rigid-as possible) method [55], it is possible to synthesize a cloth model even in a non-rectangular shape as tested by Chen et al. [27, 28].

## Implementation

To create the results of this study, we used a computer equipped with Intel Core i7–7700k CPU, 32GB RAM, NVidia GTX 1080Ti GPU, and the following SR model was used (Fig 7): We use the residual complementation method by adding the feature map after the first convolution operation is completed to the value obtained through the subsequent two convolutions. This process mitigates the error lost during the convolution operation through residual

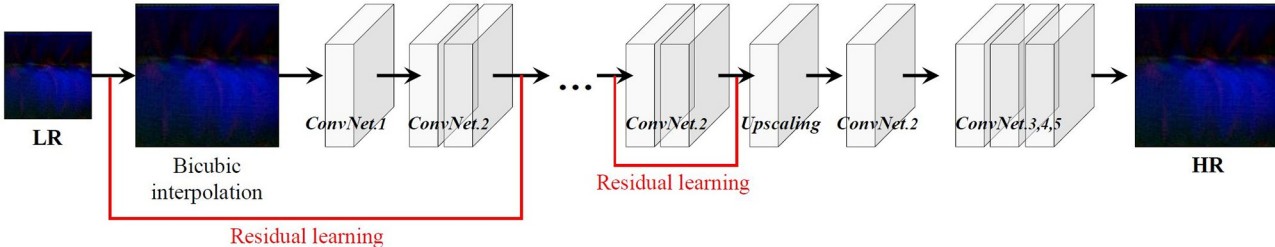

**Fig 7. VGG19 neural network structure (red: Residual process).**

complementation (This process based on residual is only allowed for GeometryNet). We repeated this process 10 times, and since it goes through 2 convolutions per cycle, a total of 20 convolution operations are performed. At first, the value after the first convolution is added, but after that, the previous value is repeatedly added. Then, the size is doubled through upscaling, and after 4 convolution operations are performed, the entire process is finished. Since BoundaryNet and EnhanceNet do not use the residual approach, the above process is omitted in these two networks.

As training data, the CIFAR-10 and COCO 2017 datasets were used together with the geometry images of cloth meshes obtained by physically based simulation. The SR scale was doubled, the batch size was 32, and the learning rate was 0.0001 for a total of 100,000 iterations, and Adam was used as the optimizer.

## Results and discussion

In this paper, we proposed a framework that can efficiently model cloth by converting 3D cloth meshes into 2D geometry images and then applying the SR technique. In this process, a cotangent weight-based Laplacian operator and anisotropic kernel were used to alleviate the surface noise and shrinkage problems appearing in ConvNet. Unlike recent techniques that were difficult to apply in complex cloth scenes due to noise and shrinkage problems, the proposed method produced good results even in complex scenes with cloth twisting.

Fig 8 is a scene with a cloth flapping back and forth, the input cloth data used has a $32 \times 32$ resolution (Fig 8A) and we upscaled it by 2x. As shown in Fig 8B, in the previous method, distortion is evident at the boundary when the cloth moves rapidly. Furthermore, the distortion appeared even when the cloth was moved while the top of the cloth was fixed, indicating that simply duplicating the boundary pixels could not solve this problem. On the other hand, our method synthesized the cloth surfaces without distortion (Fig 8C and 8D).

To test our method in various environments, we created a scene of cloth surfaces colliding with obstacles (Fig 9). The results of our method show that the distortion is greatly mitigated, as in the previous result (Fig 9C and 9D). Compared with Fig 8, the distortion was larger in Fig 8 where the external force was strongly applied. On the other hand, since the force applied to the cloth is reduced due to the repulsive force caused by the sphere and collision, the noise is relatively weak, but it is still considered a critical error from the VFX point of view (Fig 9B).

Fig 10 is a scene where cloth simulation was performed after fixing 4 corner vertices, and all of them are results of using our method. In particular, this scene is an experiment to observe the effectiveness of the BoundaryNet proposed above. In each row, the left subfigure is the input data, and unlike the previous results, since the corners are fixed, the cloth surfaces that sag in a U-shape can be observed. Fig 10A shows a different form of distortion than before. In Figs 8 and 9, most of the distortion occurred at the corner, whereas in Fig 10A, distortion

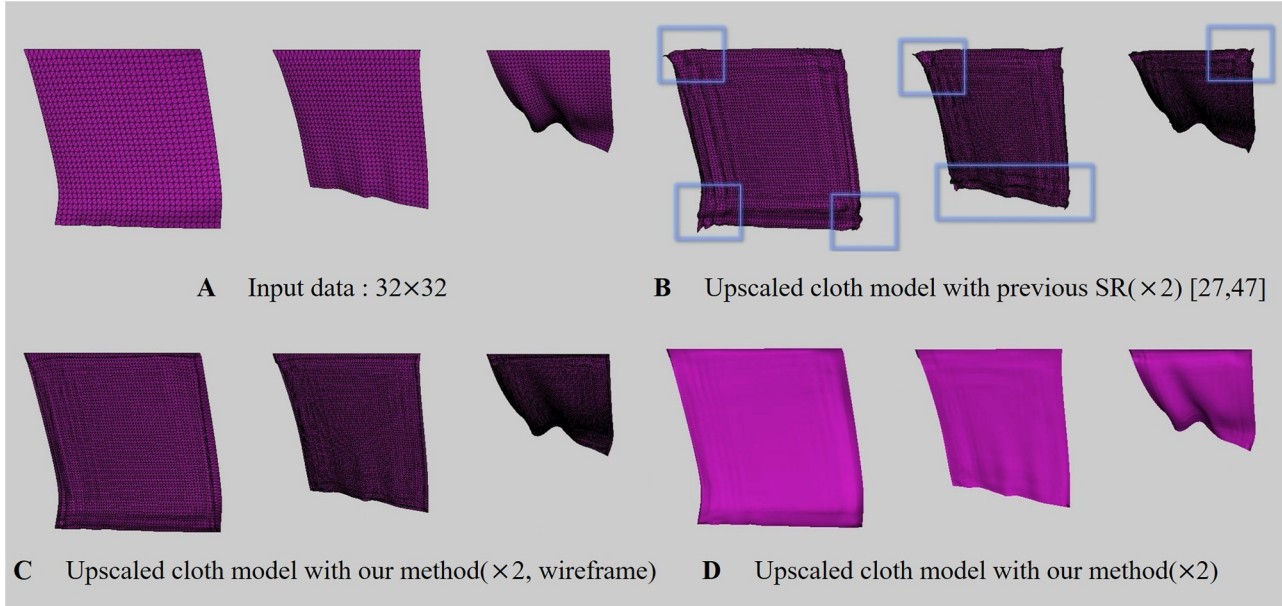

**Fig 8. Cloth model flapping back and forth (box: Distortion region).** The results are presented in S1 Video.

occurred at the border line. In Figs 8 and 9, the entire border line is fixed and receives almost no force, but in Fig 10A, the force is clearly transmitted, so distortion occurs. Our method *without* BoundaryNet also has weaker distortion than previous methods [27, 28] (Fig 10A-right), but when BoundaryNet is applied to our method, it definitely produced good results (Fig 10A-middle). The same results were also found in other frames (Fig 10A ∼ 10C).

Cloth simulation is frequently used not only in VFX and virtual fashion fields, but also in various game effects. If the previously shown U-shaped surfaces cannot be synthesized into HR cloth surfaces, this will only be usable in limited scenes, and it will be even more difficult to use for effects such as tearing cloth. In general, U-shaped surfaces appear in parts sagged

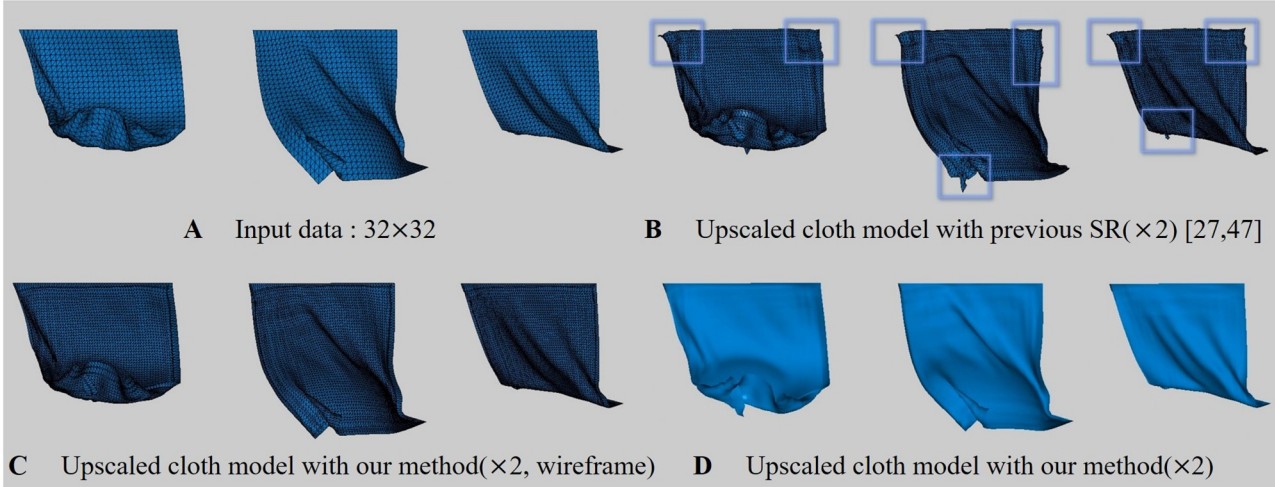

**Fig 9. Cloth model flapping back and forth with collision (box: Distortion region).** The results are presented in S1 Video.

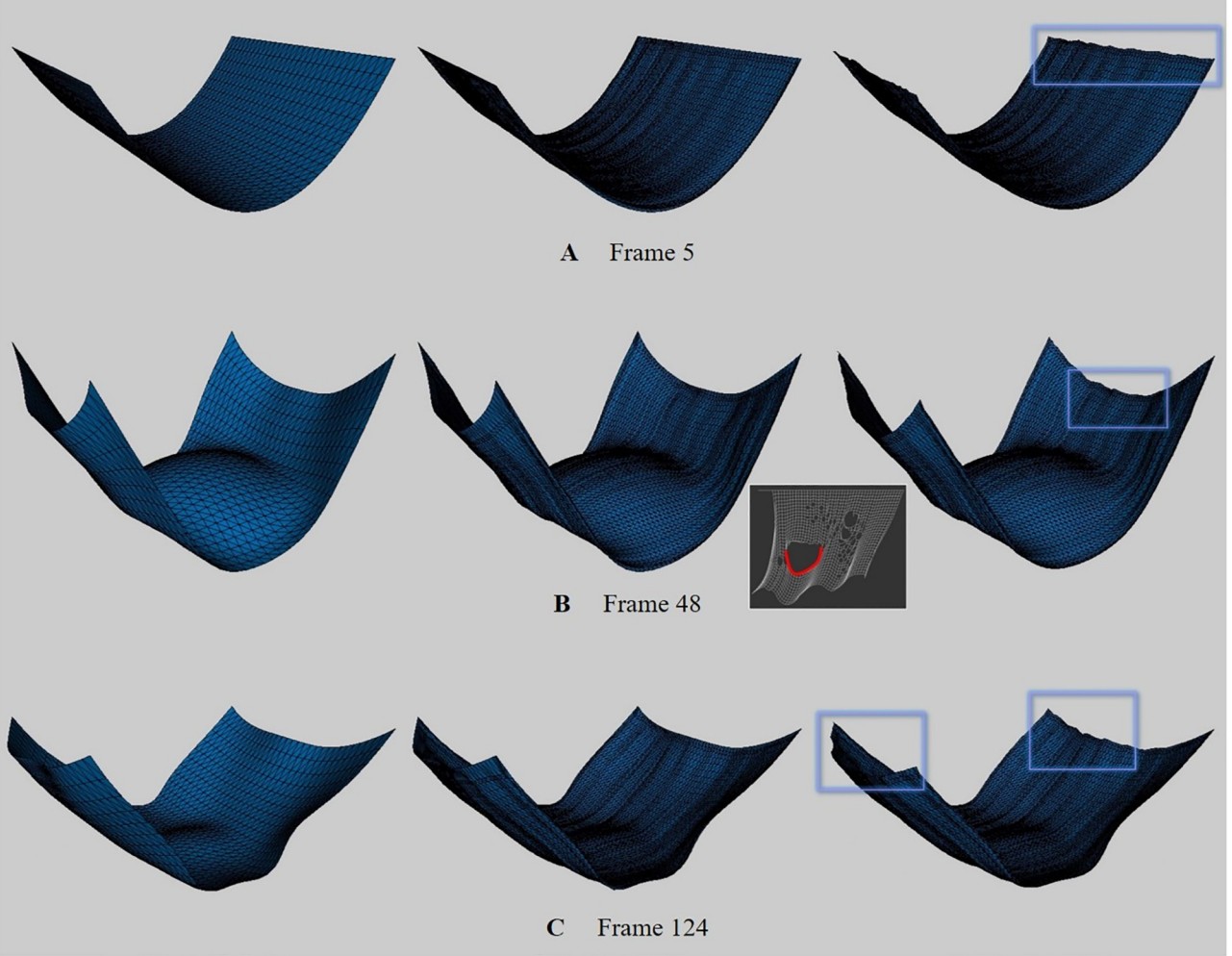

**Fig 10. Cloth falling on top of sphere (box: Distortion region).** In the result of each row, the left image is the input data, the middle one is our method, and the right one is our method *without* BoundaryNet. The results are presented in S1 Video.

down by fixed points, but they also appear frequently in tearing effects as in the inset image of Fig 10B. Our method is expected to be highly effective in such scenes.

Fig 11 is a scene that twists the cloth surfaces. The input cloth data used has a $32 \times 10$ resolution (Fig 11A) and we upscaled it by 2x. In the previous methods [27, 28], it can be seen that distortion appears in the regions from **A** to **F** (Fig 11D): In **A**, the vicinity of the boundary was distortion by twisting force, and in **B** and **C**, surfaces were shrinked. In particular, shrinkage and noise problems were prominent in twisting motion. **E** shows that our method preserves the surface shape well when compared with the input data. In the previous methods, it can be seen that the detail is somewhat lost compared to the original data. **F** is a sharply twisted tip, and the previous approach shows the cloth surfaces that are shrinking compared to the sharp patterns of the original surfaces, but our method shows the sharp surfaces well.

Fig 12 shows the results of experiments using various elastic materials. As with the previous results, cloth surfaces were synthesized without boundary shrinkage and noise problems in various materials. Table 1 shows the simulation environment used in this paper.

Fig 13 is the result of SR experiments on cloth deformation without external forces, and gravity was not used to generate this scene. As demonstrated previously, we can produce a

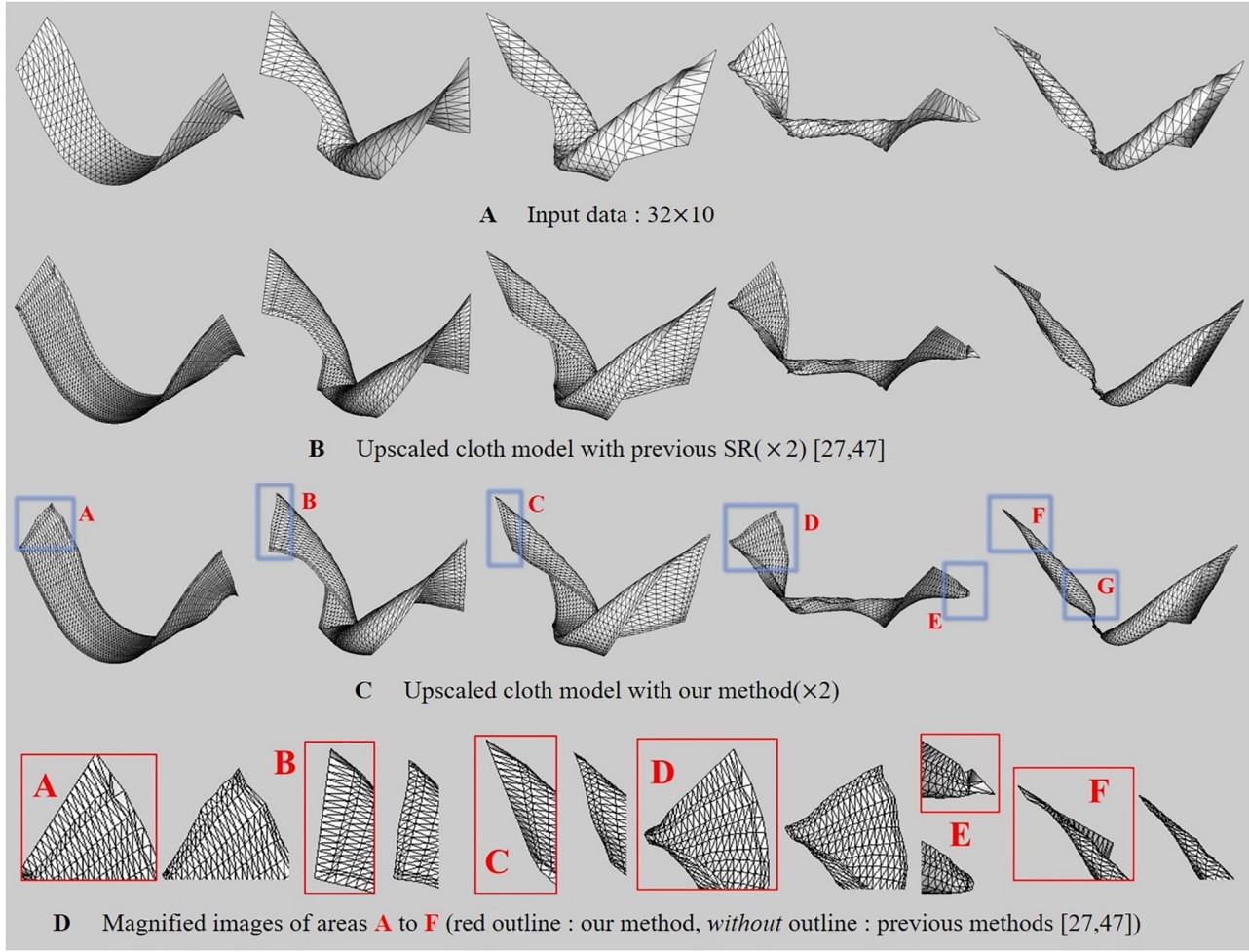

**Fig 11. Twisted cloth model (box: Distortion region).** The results are presented in S1 Video.

stable cloth SR near the boundary despite the large deformation. Also, the upscaled resolution of the cloth was set at random, and distortion did not appear in the over-twisted area as shown in Fig 13B.

During the initial design phase of the algorithm, we thought about which one to choose between GAN and ConvNet. Due to the convolution filter, the detail of the feature map created by ConvNet is rather smoothed in comparison to GAN. However, because each pixel corresponds to a single vertex location in our method, the ConvNet with smoothed style was preferable to the GAN, which gives a detailed feature map via a nonlinear filter. As a result, we believe it will perform well in Xception [56], a ConvNet with 71 layers.

Similarly to how increasing the upsampling resolution does not always improve the results of image SR, our technique has limitations. While the nonlinear filter improves the quality of image and geometry SR, noise is frequently introduced by over-SR when the resolution is set too high. In our method, increasing the resolution by more than 3 times occasionally resulted in awkward upsampling results, and over-SR prevented wrinkle enhancement from performing properly. Experiments in a variety of scenes produced the most stable and satisfactory results at twice the resolution.

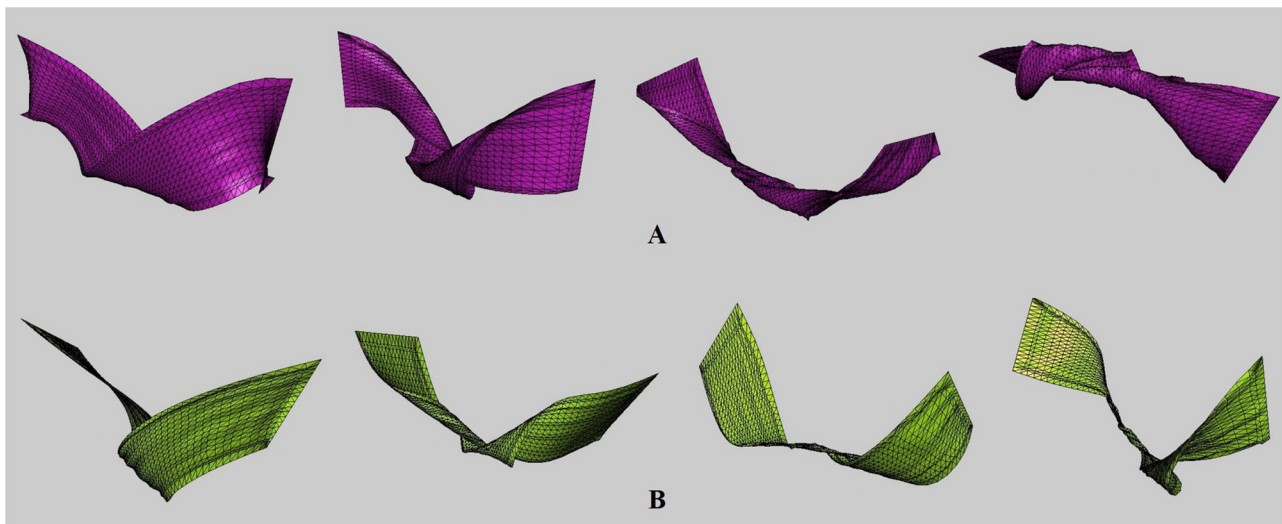

**Fig 12. Twisted cloth model with our method(×2).** The results are presented in S1 Video.

**Table 1. Environment settings for each example scene.**

| Figure | Num. of triangles (LR→HR) | Size of geometry image (LR→HR) | Scale factor |
|---|---|---|---|
| Fig 8 | 1,922→7,938 | 32×32 → 64×64 | ×2 |
| Fig 9 | 1,922→7,938 | 32×32 → 64×64 | ×2 |
| Fig 10 | 1,922→7,938 | 32×32 → 64×64 | ×2 |
| Fig 11 | 558→2,394 | 32×10 → 64×20 | ×2 |
| Fig 12 | 558→2,394 | 32×10 → 64×20 | ×2 |

Prior to sampling, a parametrization step is required to convert non-rectangular meshes to rectangular structures. There are several techniques to parametrization, including the widely used As-Rigid-As-Possible (ARAP) method [55].

## Conclusions and future work

In this paper, AnisoCBConvNet, a new neural networks method that expresses HR cloth surfaces by converting LR cloth surfaces into geometry images, was described. We modeled three

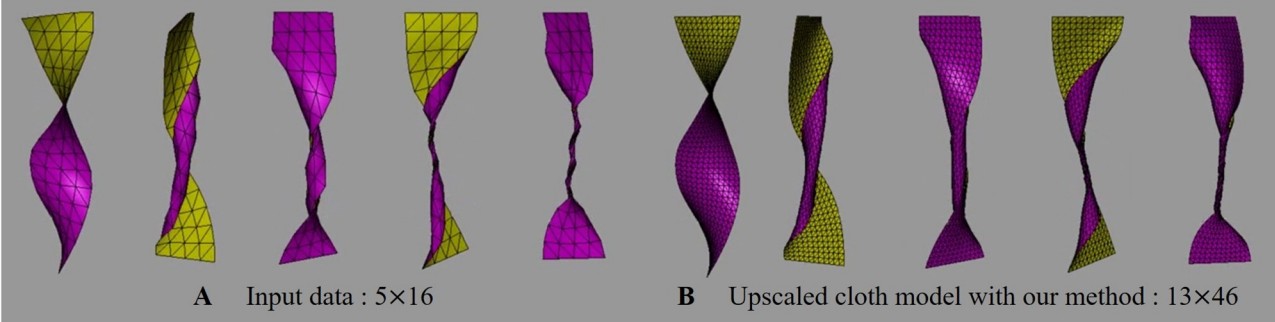

**Fig 13. Twisted cloth model with our method.** Gravity was not applied to this scene.

networks (GeometryNet, BoundaryNet, EnhanceNet) to alleviate the problems that occur when reconstructing 3D cloth surfaces, and introduced a method for compositing the results. Unlike the existing methods in which distortion occurs near the boundary when the cloth surfaces are greatly deformed, our method clearly expresses the HR cloth surfaces. In addition, since U-shaped surfaces are stably expressed, it can be used not only for general cloth simulations but also for tearing effects.

In this paper, we propose a new network architecture that upsamples cloth simulation using VGG19. Because VGG19 network is used in a variety of applications using ConvNet, it has also been adopted in this paper. Our solution is not network-specific, and because it is stable in VGG19, it may be used and expanded to other ConvNet-based approaches. We will attempt to improve the algorithm in the future by applying it to the model you mentioned, the Xception Network [56].

Nevertheless, in our method, we observed that noise appeared on the surfaces in the plasticity material model. Unlike elastic materials, where there is no significant change in the boundary shape, plasticity materials either retain their stretched shape or often have different boundaries from their original shape, so a different approach is required to apply them. Our method did not consider the plasticity material because the framework was designed assuming a general cloth model. In the future, we plan to study networks that can apply SR to cloth surfaces with plasticity material.

## Supporting information

**S1 Video. Supplementary result data.** Related to Figs 8 ∼ 13.
(AVI)

## Author Contributions

**Conceptualization:** Jong-Hyun Kim.

**Data curation:** Jung Lee.

**Methodology:** Sun-Jeong Kim.

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
