## [Decision Letter · Decision Letter 0]

29 Oct 2021

PONE-D-21-28016Geometry Image Super-Resolution with AnisoCBConvNet Architecture for Efficient Cloth ModelingPLOS ONE

Dear Dr. Kim,

Thank you for submitting your manuscript to PLOS ONE. After careful consideration, we feel that it has merit but does not fully meet PLOS ONE’s publication criteria as it currently stands. Therefore, we invite you to submit a revised version of the manuscript that addresses the points raised during the review process.

We look forward to receiving your revised manuscript.

Kind regards,

Sen Xiang

Academic Editor

PLOS ONE

Journal Requirements:

Reviewers' comments:

Reviewer's Responses to Questions

**Comments to the Author**

1. Is the manuscript technically sound, and do the data support the conclusions?

Reviewer #1: Yes

Reviewer #2: Partly

2. Has the statistical analysis been performed appropriately and rigorously? 

Reviewer #1: Yes

Reviewer #2: Yes

3. Have the authors made all data underlying the findings in their manuscript fully available?

Reviewer #1: Yes

Reviewer #2: No

4. Is the manuscript presented in an intelligible fashion and written in standard English?

Reviewer #1: No

Reviewer #2: Yes

5. Review Comments to the Author

Reviewer #1: 1) Why authors are used VGG19, because already best models are available like Xception Network, MobileNetwork etc.

2) If VGG19 response is better, plz shows the comparable results with Xception and Inception model

3)Plz add latest referenc e2020 and 2021 in your article.

4) Why used the bicubic interpolation in Figure 7 to upscale the LR image to HR image, because interpolation is not designed for this purposes, all cuurent research replaced interpolation with deconvolution layer or subpixel layer to upscale the LR image to HR image.

5) if authors also calculated results on 3x and 4x scale it is better.

Reviewer #2: Contribution

This paper presents an anisotropic constrained-boundary convolutional neural network to enhance wrinkles in cloth animation.

The main technical novelty is to anisotropically control generated boundary points to further alleviate the oscillation problems in wrinkle enhancement.

References

After reading the section of related work, this section is detailed and comprehensive, but there are still some references that are closely related to this submission, they should be discussed carefully. I list them as following:

- Meng Zhang, Duygu Ceylan, Tuanfeng Wang, Niloy J. Mitra, Dynamic Neural Garments, arXiv 2021

- Tuanfeng Y. Wang, Tianjia Shao, Kai Fu, Niloy J. Mitra, Learning an intrinsic garment space for interactive authoring of garment animation, ACM Transactions on Graphics (TOG), vol. 38, no. 6, p. 220, 2019.

- Lan Chen, Lin Gao, Jie Yang, Shibiao Xu, Juntao Ye, Xiaopeng Zhang, and Yu-Kun Lai, Deep deformation detail synthesis for thin shell models, arXiv 2021.

- Meng Zhang, Tuanfeng Wang, Duygu Ceylan, Niloy J. Mitra, Deep Detail Enhancement for Any Garment, Eurographics 2021

Technical correctness

Yes, it is technically correct. Some experiments and explanations, however, should be added.S

1. The authors mentioned the flickering and over-shrinkaged problems at first but they seems not solve all these problems but boundary oscillation. I would like to see more results solving these two problems.

2. One of the main contribution is the anisotropic constrained-boundary network to generate stable boundaries.

However, this blur boundary problem is due to the convolutional operation at the image boundaries. It is not the special problem in geometry images but a common issue in image super-resolution.

You can test the SOTA SR model. In image SR field, researches often calculate the PSNR value after cropping boundaries.

Besides, in mesh generation, Chen et.al used padding operations to solve the boundary problems. I would like to see more convincing results of your contribution.

3. You mentioned that "It does not work properly ... but only properly in SRResNet ...", but your method is also a residual superresolution network. Is there any difference? And I think this network is not the boundary solution.

4. The paper only use square and rectangular meshes as examples. Please give more results to demonstrate the correctness of your boundary network.

Writing and organization

The writing and organization in this paper needs to be improved. The authors use long sentences and the logic between sentences needs to be improved.

For example:

- Page1, "an anisotropic ... networks ... technique" -> "an anisotropic ... network" .

- Page1, "As a training set to be provided as an input to the neural network" too long

- Page1 "details including wrinkles" repeated.

- Fig 2. Which ANN did you use ? Please give citations.

- spelling issues line 246 "ground-true" -> "ground-truth"

6. PLOS authors have the option to publish the peer review history of their article (what does this mean?). If published, this will include your full peer review and any attached files.

Reviewer #1: No

Reviewer #2: **Yes: **Lan Chen

---

## [Author Response · Author response to Decision Letter 0]

17 Mar 2022

Attach Files (Response_to_Reviewers.docx file)

---

## [Decision Letter · Decision Letter 1]

31 May 2022

PONE-D-21-28016R1Geometry Image Super-Resolution with AnisoCBConvNet Architecture for Efficient Cloth ModelingPLOS ONE

Dear Dr. Kim,

Thank you for submitting your manuscript to PLOS ONE. After careful consideration, we feel that it has merit but does not fully meet PLOS ONE’s publication criteria as it currently stands. Therefore, we invite you to submit a revised version of the manuscript that addresses the points raised during the review process.

We look forward to receiving your revised manuscript.

Kind regards,

Sen Xiang

Academic Editor

PLOS ONE

Journal Requirements:

Reviewers' comments:

Reviewer's Responses to Questions

**Comments to the Author**

1. If the authors have adequately addressed your comments raised in a previous round of review and you feel that this manuscript is now acceptable for publication, you may indicate that here to bypass the “Comments to the Author” section, enter your conflict of interest statement in the “Confidential to Editor” section, and submit your "Accept" recommendation.

Reviewer #1: All comments have been addressed

Reviewer #3: (No Response)

2. Is the manuscript technically sound, and do the data support the conclusions?

Reviewer #1: Yes

Reviewer #3: (No Response)

3. Has the statistical analysis been performed appropriately and rigorously? 

Reviewer #1: I Don't Know

Reviewer #3: (No Response)

4. Have the authors made all data underlying the findings in their manuscript fully available?

Reviewer #1: Yes

Reviewer #3: (No Response)

5. Is the manuscript presented in an intelligible fashion and written in standard English?

Reviewer #1: Yes

Reviewer #3: (No Response)

6. Review Comments to the Author

Reviewer #1: (No Response)

Reviewer #3: The writing in this paper needs to be improved.

For example:

Page6, "To alleviate the problems of ... when reconstructed from ... to ... " lack of sentence components

7. PLOS authors have the option to publish the peer review history of their article (what does this mean?). If published, this will include your full peer review and any attached files.

Reviewer #1: No

Reviewer #3: No

---

## [Author Response · Author response to Decision Letter 1]

7 Jul 2022

We have attached the file (Attach Files-Response_to_Reviewers.docx .docx).

---

## [Decision Letter · Decision Letter 2]

20 Jul 2022

Geometry Image Super-Resolution with AnisoCBConvNet Architecture for Efficient Cloth Modeling

PONE-D-21-28016R2

Dear Dr. Lee,

We’re pleased to inform you that your manuscript has been judged scientifically suitable for publication and will be formally accepted for publication once it meets all outstanding technical requirements.

Kind regards,

Sen Xiang

Academic Editor

PLOS ONE

Additional Editor Comments (optional):

Reviewers' comments:

Reviewer's Responses to Questions

**Comments to the Author**

1. If the authors have adequately addressed your comments raised in a previous round of review and you feel that this manuscript is now acceptable for publication, you may indicate that here to bypass the “Comments to the Author” section, enter your conflict of interest statement in the “Confidential to Editor” section, and submit your "Accept" recommendation.

Reviewer #1: All comments have been addressed

2. Is the manuscript technically sound, and do the data support the conclusions?

Reviewer #1: Yes

3. Has the statistical analysis been performed appropriately and rigorously? 

Reviewer #1: Yes

4. Have the authors made all data underlying the findings in their manuscript fully available?

Reviewer #1: Yes

5. Is the manuscript presented in an intelligible fashion and written in standard English?

Reviewer #1: Yes

6. Review Comments to the Author

Reviewer #1: (No Response)

7. PLOS authors have the option to publish the peer review history of their article (what does this mean?). If published, this will include your full peer review and any attached files.

Reviewer #1: No

---

## [Editor Report · Acceptance letter]

12 Aug 2022

PONE-D-21-28016R2 

Geometry Image Super-Resolution with AnisoCBConvNet Architecture for Efficient Cloth Modeling 

Dear Dr. Lee:

I'm pleased to inform you that your manuscript has been deemed suitable for publication in PLOS ONE. Congratulations! Your manuscript is now with our production department. 

Kind regards, 

on behalf of

Dr. Sen Xiang 

Academic Editor

PLOS ONE